# The Gender Impact Assessment among Healthcare Workers in the SARS-CoV-2 Vaccination—An Analysis of Serological Response and Side Effects

**DOI:** 10.3390/vaccines9050522

**Published:** 2021-05-18

**Authors:** Chiara Di Resta, Davide Ferrari, Marco Viganò, Matteo Moro, Eleonora Sabetta, Massimo Minerva, Alberto Ambrosio, Massimo Locatelli, Rossella Tomaiuolo

**Affiliations:** 1IRCCS San Raffaele Hospital, 20132 Milan, Italy; moro.matteo@hsr.it (M.M.); ambrosio.alberto@hsr.it (A.A.); locatelli.massimo@hsr.it (M.L.); 2University of Parma, 43121 Parma, Italy; davide.ferrari@unipr.it; 3IRCCS Galeazzi Orthopaedic Institute, 20161 Milan, Italy; marco.vigano@grupposandonato.it; 4Vita-Salute San Raffaele University, 20132 Milan, Italy; e.sabetta@studenti.unisr.it (E.S.); m.minerva3@studenti.unisr.it (M.M.); tomaiuolo.rossella@hsr.it (R.T.)

**Keywords:** COVID-19, side effects, vaccination coverage, gender impact assessment

## Abstract

Healthcare professionals are considered to be at high risk of exposure and spread of SARS-CoV-2, and have therefore been considered a priority group in COVID-19 vaccination campaign strategies. However, it must be assumed that the immune response is influenced by numerous factors, including sex and gender. The analysis of these factors is an impact element for stratifying the population and targeting the vaccination strategy. Therefore, a large cohort of healthcare workers participating in the Italian vaccination campaign against SARS-CoV-2 has been studied to establish the impact of sex and gender on vaccination coverage using the Gender Impact Assessment approach. This study shows a significant difference in the antibody titers among different age and sex groups, with a clear decreasing trend in antibody titers in the older age groups. Overall, the serological values were significantly higher in females; the reported side effects are more frequent in females than in males. Therefore, disaggregated data point out how the evaluation of gender factors could be essential in COVID-19 vaccination strategies. On this biomedical and social basis, suggestions are provided to improve the SARS-CoV-2 vaccination campaign in healthcare professionals. Still, they could be adapted to other categories and contexts.

## 1. Introduction

Healthcare workers are one of the most at risk groups for being exposed to SARS-CoV-2 infection and transmitting it, particularly among patients [1]. Thus, in the limited COVID-19 vaccine supply, they were included as a priority group for vaccination [2]. However, it is crucial to evaluate COVID-19 vaccine coverage since many factors influence the antibody response [3].

The SARS-CoV-2 infection causes different immune responses in men and women in prevalence, intensity, and outcome [4], including in the cases of natural infection and vaccination [5,6]. These differences were already known in the literature for other viral infections [7] and in general for the immune response [8,9]. Underlying these differences is a combination of nature and nurture [10,11,12].

In particular, sex hormones differentially modulate immune responses [13]. In the male, testosterone has a suppressive effect on the immune functions [14]; moreover, androgens exert an inhibitory effect on the differentiation of Th1, with consequently reduced production of IFN-γ, explaining the enhanced susceptibility to viral infections in males than in females [14,15]. In females, estrogens have different effects on the immune system: (1) the physiological concentration of estrogens stimulates a humoral response to viral infections by inducing higher levels of antibodies and activating antibody-producing cells; (2) low concentrations of estrogens induce monocyte differentiation into inflammatory dendritic cells with consequent high production of IL-4 and IFN-α; and (3) high concentrations of estrogen have inhibitory activities on innate and proinflammatory immune responses [16,17]. In addition, females show a better response to vaccination [18,19], however, side effects are significantly more frequent [20].

In addition, gender (the social construction of femininity and masculinity, which includes sociocultural and psychological aspects) seems to play a significant role in determining the immune response to SARS-CoV-2. Indeed, health opportunities and risks vary according to social, economic, environmental, and cultural influences [21]. Experience from past outbreaks shows the importance of incorporating a gender analysis to improve health interventions’ effectiveness and to promote gender and health equity goals [22]. However, despite several efforts to examine gender on health outcomes [22,23,24], it is rarely a main consideration because the gender assessment tools are not commonly used for analyzing gender influence on health outcomes in the real world [25].

Therefore, data disaggregated by sex and gender are essential to carry out effective interventions against SARS-CoV-2 from a biomedical and social perspective [26].

This study, aiming to improve the efficiency of the SARS-CoV-2 vaccination campaign, provides sex and gender-based information to support the management of vaccination campaigns against SARS-CoV-2. At this scope, the impact of sex and gender on immune response against SARS-CoV-2 was evaluated in a large cohort of health workers adhering to the Italian vaccination campaign against SARS-CoV-2.

## 2. Materials and Methods

### 2.1. Study Design

The impacts of sex and gender on the immune response against SARS-CoV-2 were tested by the Gender Impact Assessment (GIA) method [27], using antibody titer and side effects as indicators. The study was conducted on a population of 3318 hospital-based healthcare workers, considering sex and age, as described below, starting from January 2021 to the middle of February 2021.

Moreover, healthcare professionals with a prior COVID-19 clinical diagnosis were not included, according to the Medical Management of the hospital’s recommendations. The antibody titer was obtained in all the study groups with the Roche methods. The results are comparable; the vaccine administered was always the BNT162b2 mRNA COVID-19 Vaccine, thus the response and side effects depend exclusively on the recipient.

### 2.2. Gender Impact Assessment (GIA)

GIA is a stepwise process, as reported below in Table 1. To assess the gender impact of any policy or activity, first (step 1) it is necessary to define the context (identification of the problem and its impact on the society) and to make explicit the objectives set and the indicators used to track and monitor inequalities. Then (step 2), it is required to explain the importance of having introduced the gender determinant, identifying the gender dynamics and the related direct impacts (i.e., access to resources, payment methods/costs, etc.) and the indirect gender impacts (intermediate access to resources, services, institutions, structures, etc.). At this point (step 3), it is needed to identify prospective gender impacts, for example, gender stereotypes that influence knowledge and behaviour, i.e., hierarchical positioning that generates social, cultural, and economic privileges; unequal use and access to resources; and unfair and unbalanced representation. Finally, in steps 4 and 5, the impact of gender is assessed (the harmful implications of gender bias, which aspects reinforce or reduce inequalities, and which factors promote equality over the status quo), and the recommendations for design adjustments are defined (suggesting how to reduce disparities and promote gender equality, to revisit predicted negative impacts and to develop strategies to transform them into positive effects).

### 2.3. Population

The study population was composed of 3318 healthcare workers (including physicians, nurses, and laboratory personnel) at IRCCS San Raffaele Hospital during the Italian COVID-19 vaccination campaign.

According to the approved protocol (CE:199/INT/2020) of the Institutional Ethical Review Board, written informed consent was obtained from all the participants. Health workers included all staff categories working in the hospitals who were eligible for vaccination with no contraindication to receiving the COVID-19 vaccine. The population was divided into the age groups of <48, 48–52, and >52 for women (based on the decrease of estrogen with menopause) [28], and <40, 40–60, and >60 for men (based on the gradual decline level of testosterone with ageing in men) [29]. The same subdivision was maintained to assess the occurrence of side effects due to the vaccine. Clinical data of chronic diseases affecting the enrolled participants are not available.

The distribution in terms of sex and age of the whole healthcare workers at IRCCS San Raffaele Hospital is described in Appendix A.

### 2.4. Data Sources

Serial blood samples were collected for serological evaluation at three time points (Figure 1): baseline serum sample (T_0_), at 21st day (T_1_), and 42nd day (T_2_).

At T_0_, serum samples were tested by the Elecsys (Roche, Basel, Switzerland), Anti-SARS-CoV-2 assay on the COBAS 601 platform (Roche, Basel, Switzerland) for the determination of antibodies (total immunoglobulin IgT) specific for the viral SARS-CoV-2 nucleocapsid (N) protein [30]. As reported in the manufacturer’s datasheet (ref: 09289267501V0.6), the result is given as a cutoff index (COI) as well as in the form of qualitative results: for COI < 1.0, the sample is nonreactive and negative for anti-SARS-CoV-2 antibodies; for COI > 1.0, the sample is reactive and positive for anti-SARS-CoV-2 antibodies. The manufacturer indicated specificity (95% CI) of 99.80% and sensitivity (95% CI) of 99.5 ≥ 14 days post-PCR confirmation. The estimated performance reported by the FDA [31] was 100% sensitivity, 99.8% specificity, 96.5% PPV, and 100% NPV (both at 5% prevalence).

At T_1_ and T_2_, samples were tested by the Elecsys Anti-SARS-CoV-2 assay on COBAS 601 platform (Roche, Basel, Switzerland), targeted on IgT against the receptor-binding domain (RBD) of the viral spike protein (S-protein). As reported in the manufacturer’s datasheet (ref: 09289267501), the SARS-CoV-2-S test has a signal range ranging from 0.4 to 250 U/mL. The manufacturer indicated <0.80 U/mL as negative for anti-SARS-CoV-2-S and ≥0.80 U/mL as positive for anti-SARS-CoV-2-S; values above the measuring range are reported as ≥2500 U/mL for 10-fold diluted samples. The estimated performance reported by the FDA [31] was 96.6% sensitivity, 100% specificity, 99.7% PPV, and 99.8% NPV (both at 5% prevalence).

To evaluate the side effects due to the COVID-19 vaccine, an anonymized online survey was proposed via professional email contacts and completed on the SurveyMonkey platform (SurveyMonkey^®^-SurveyMonkey.com). The survey was focused on the incidence of (a) localized reactions (pain, swelling, redness, swelling at the injection site); (b) systemic reactions (fever, tiredness/malaise, chills, headache, vomiting/nausea, diarrhea, body aches, swollen lymph nodes, dizziness/confusion); (c) allergic reactions (widespread itching, rash other than the injection site, asthma, throat tightness, anaphylaxis).

### 2.5. Statistical Analysis

The analyses were performed using R software v4.0.3 (R Core Team, Wien, Austria). Data distribution was assessed using the Shapiro Wilk test. To evaluate the significance of any difference obtained between the tested groups, a nonparametric test was used. Data are presented as median and interquartile ranges (IQR). Then, the Mann–Whitney test was applied when comparing two groups, while the Kruskall–Wallis test with Dunn’s post hoc test (with Bonferroni’s correction) was used to compare three or more groups. Correlation analyses were performed using Spearman’s method. General linear models were used to assess the contemporary influence of sex and age on the antibody titer on serum. Chi-square test and proportion trend tests were used to evaluate the differences among groups for categorical variables. *p*-Values < 0.05 were considered statistically significant.

## 3. Results

### 3.1. Assessment of the Sex and Gender Impact

The results obtained by the Gender Impact Assessment are reported in Table 1 and clarified below.

The context is the efficiency of the vaccination campaign. Female and male healthcare workers differently experienced COVID-19 and exhibited health-seeking behaviors and outcomes due to sex and gender [21]. The indicators used to track and monitor sex and gender differences were the antibody titer against SARS-CoV-2 before and after vaccination and the side effects of vaccination [20].

The exposure to the virus [32] and access to the vaccine [33] are the dynamics to consider. These dynamics are based on gender stereotypes (i.e., women are considered more suitable for care and assistance) affecting the behaviors, as proved, for example, in China and Europe, where most healthcare personnel are women (90% and 76%, respectively) [34]. In this study, the San Raffaele Hospital healthcare workers reflect this, with 64.4% of the healthcare staff being female and 35.6% being male. Furthermore, unequal conditions are noticed between hospital-based healthcare personnel vs. informal caregivers and healthcare personnel outside of hospitals. Indeed, the vaccine is strictly recommended for all front-line health personnel, but the institutions prioritise access to based-hospital healthcare workers [21]. This aspect is confirmed in this study. Indeed, the vaccination campaign has been organised explicitly by hospital Medical Management in different phases, prioritizing front-line professionals and older people who have not had a previous clinical diagnosis of COVID-19, even because of the limited availability of vaccines.

The harmful impact caused by the gender differences was that women are more likely to contact the virus, given their predominant roles as front-line healthcare workers and caregivers [35]. Therefore, providing the vaccine to all individuals at comparable risk in healthcare could reduce inequalities.

### 3.2. Assessment of the Immune Response against SARS-CoV-2

Serological tests were used to track and monitor the immune response against SARS-CoV-2.

#### 3.2.1. Serological Evaluation at T_0_

Nine percent of cases (5.4% females and 3.4% males) were positive at T_0_, indicating a natural immune response to SARS-CoV-2 in these subjects. Appendix A shows the median (±IQR) of serological values obtained in this group, disaggregated by sex and age categories, as described in the Methods section. These subjects showed a high reaction after both vaccine doses, having median serological values above the instrumental measuring range (≥2500 U/mL for 10-fold diluted samples), as reported in Figure 2.

#### 3.2.2. Serological Evaluation at T_1_

Of the seronegative subjects (i.e., without natural immunity against SARS-CoV-2) 98.2% had reactivity in response to the first vaccine dose, while only 55 subjects (1.8% of T_0_ seronegative individuals) showed an antibody titer below the cutoff threshold value, with a statistically significant difference between males (2.6%) and females (1.4%) (*p* = 0.026). A significantly increased number of nonreactive individuals was observed in older age groups, both in males (*p* < 0.0001) and in females (*p* = 0.0002) (Figure 3).

The aggregated analysis of the reactive subjects showed a median value of 43.4 U/mL (IQR: 15.9–110.0 U/mL) (Figure 4A). However, considering the disaggregated data by sex and age groups, the serological values highlighted an inversely proportional correlation between age and antibody titer (Appendix A) in both males and females (Figure 4B). Specifically, in males, a significant reduction of antibody titers was present among all the age categories (*p* < 0.0001); in females, a significant difference was observed among the <48 yr and >52 yr groups (*p* < 0.0001).

#### 3.2.3. Serological Evaluation at T_2_

These serological tests were carried out to evaluate the vaccination’s immune response. Aggregated data (Figure 4A) showed a vaccination’s immune response in 99.92% of subjects and a median value of 1653 U/mL (IQR: 871.5–2500 U/mL). Only 0.08% (n = 3) of the vaccinated subjects showed an antibody titer below the cutoff, and they were exclusively men.

Considering the disaggregated data among the responder subjects, the antibody titers were influenced by sex and age (Figure 4B), as in T_1_ data (Appendix A). Specifically, a statistically significant difference was observed among <40 yr males vs. the >60 yr (*p* = 0.01) and the 40–60 yr (*p* = 0.036) age groups. Furthermore, the younger females (<48 yr) showed a significantly higher antibody titer than the 48–52 yr subjects (*p* = 0.014) and the >52 yr ones (*p* < 0.001). Interestingly, the overall serological values were significantly higher in females than in males (*p* = 0.0002).

### 3.3. Assessment of the Side Effects Due to the COVID-19 Vaccine

An online survey was conducted among the enrolled healthcare workers to assess the tolerability and reactogenicity of COVID-19 vaccines. A total of 2482 online surveys were completed: 1093 respondents after the first vaccine dose (19%) and 1389 (29%) after the second vaccine dose. Overall, 909/1093 (83.17%) had self-reported side effects after the first vaccine dose; more specifically, respondents reported experiencing more frequently localized reactions (74.29%) than systemic reactions (47.30%) or hypersensitivity reactions (2.65%). In Appendix A, the different numbers of survey replies are reported based on disaggregated data by sex and age groups.

Males and females showed similar incidences in terms of allergic reactions after the first dose (OR 1.31, 95% CI: 0.53–3.29, *p* = 0.669), while they were more frequent in the female subjects after the second dose (OR 2.28, 95% CI: 1.07–4.89, *p* = 0.030). Local reactions had a similar incidence in males and females both after the first and the second dose (*p* = 0.475 and *p* = 0.147, respectively) with an odds ratio (OR) of 1.12 (95% confidence interval: 0.82–1.52) and 1.21 (95% CI: 0.94–1.56), respectively. On the other hand, systemic reactions were significantly more frequent in females than in males after both the first and second doses (*p* < 0.001 in both cases) with an OR of 1.79 (95% CI: 1.35–2.37) and 2.04 (95% CI: 1.58–2.36) respectively (Figure 5).

In general, the second vaccine dose was associated with a higher incidence of systemic reactions, reported more in women (82.13%) than in men (71.11%); again, an inversely proportional correlation was found between age and localized reactions in both males and females (Figure 6C,D).

Concerning males, considering the age groups, a significant decreasing trend was observed in association with increased age categories for both systemic (*p* = 0.001) and local reactions (*p* = 0.0002) after the first dose (Figure 6A,B). In contrast, this association was observed for a systemic reaction only after the second dose (*p* = 0.075) (Figure 6C,D). Allergic reactions had a similar incidence among age groups (*p* = 0.604 and *p* = 0.363 after the first and second dose, respectively).

A similar association between younger age and the adverse reaction was observed in female subjects. Systemic reactions were more frequent in younger females after both the first (*p* = 0.008) and second dose (*p* < 0.001), as well as local reaction (*p* < 0.001 and *p* = 0.023 after the first and second dose, respectively) (Figure 6A–D). No differences among age groups were observed concerning the allergic reactions (*p* = 0.814 after the first dose and *p* = 0.117 after the second dose).

## 4. Discussion

Various strategies to improve disease awareness, address concerns about vaccine effectiveness, and increase vaccine accessibility have been recommended [36]. The introduction of sex and gender determinants in clinical practice can contribute favourably to the management of prevention, diagnosis, and treatment strategies, making health services more effective and efficient [37,38,39,40] as these factors influence the physiological aspect and the pathological course of diseases affecting both men and women [41].

This study demonstrated that sex and gender assessment is a crucial determinant in the management of vaccine campaigns to stratify the population and organize targeted approaches to monitoring vaccination compliance. Gender Impact Assessment points out that to effectively and fairly address COVID-19 vaccination campaigns, the stereotypes and gender relationships affecting biological differences need to be considered. First of all, sex-disaggregated data of our population showed that 64.4% of healthcare workers are women [1]. To positively transform the professional risks, female healthcare workers could be included in public health strategies against COVID-19, taking into account the gained knowledge and experience. For example, moments of comparison between health professionals can be envisaged to allow the female healthcare workers to suggest bottom-up actions, ideas, and “good practices” to improve the quality of care and prevention procedures. All the initiatives and strategies adopted in healthcare contexts to strengthen prevention strategies and vaccination campaigns could increase gender equality. Indeed, the efficiency of one or more interventions to reduce inequalities among front-line health care professionals will be associated with higher compliance. Gender Impact Assessment has highlighted the inequalities of access to the vaccination campaign for informal caregivers and healthcare workers outside hospitals *vs.* hospital-based healthcare workers. Indeed, all healthcare personnel, including cleaners, catering, and waste sorting, should receive the same measures to prevent SARS-CoV-2 infection [42].

Analyzing the sex and age group is also valuable for evaluating vaccine-induced immune response against SARS-CoV-2. Indeed, after the first vaccine dose, the serological test highlights an inversely proportional correlation between age and antibody titer in both males and females (Figure 4). Interestingly, disaggregated data highlight that <48 yr females have the highest response, confirming that immune responses to viral infections may vary with sex and age. The immune and the endocrine system experiences profound changes with ageing, thereby increasing the susceptibility to infectious diseases [43] and decreasing the efficacy of vaccination [44]. Immunosenescence impacts the innate and adaptive immune system [45,46,47,48] both in females [16,49,50] and in males [51]. This could be supported by estrogens’ and androgens’ roles in promoting and suppressing the immune responses during infections and after vaccination, respectively [8]. These data should be considered for planning targeted vaccination coverage monitoring strategies; the antibody response should therefore be reassessed as a priority in male healthcare workers >60 yr and women >52 yr.

The serological test is valuable for defining the vaccine dosage. Natural seropositive individuals exhibited higher reactogenicity already after the first dose, underlining the open question of the need for a second dose in seropositive individuals [52,53]; this could increase vaccination coverage efficiency among populations in a condition of limited vaccine supply. In addition, healthcare workers with a prior COVID-19 infection were observed as having a high antibody titer (≥2500 U/mL) already after the first vaccine dose, as supported by recent studies [53].

Our data suggest that a single dose of the vaccine elicits immune responses in seropositive individuals with postvaccine antibody titers comparable to or exceeding titers found in naïve individuals who received two doses. These observations are in line with the first vaccine dose serving as a boost in naturally infected individuals, providing a rationale for updating vaccine recommendations to consider a single vaccine dose sufficient for seropositive individuals to reach immunity. A serological test that measures antibodies to the spike protein could be used to screen individuals before vaccination; this “vaccine triage” could be a way to suggest the distribution of vaccine doses based on the antibody titer. Such policies would optimize (in particular, due to the scarcity of vaccine doses and to logistic constraints) vaccination campaigns by defining vaccine inoculation priorities as a function of antibodies titer. However, it could limit the reactogenicity experienced by naturally seropositive individuals.

Compared with the first dose of the vaccine, we also found an increased incidence of side effects after the second vaccine dose (Figure 6).

It is also mandatory to include sex-disaggregated reporting of side effects of the BNT162b2 mRNA COVID-19 Vaccine. The 2482 surveys highlighted an inversely proportional correlation between age and adverse reactions. Our data confirmed that, including the COVID-19 vaccination, females more frequently develop more severe reactions due to the more intense inflammatory responses [20] (Figure 5). Therefore, these gender-based evidences point out the need for an information program and targeted monitoring. Systemic reactions were more common after the second dose than after the first dose of the vaccine, although local reactions were similar after the two doses. Finally, allergic reactions were limited in both males and females [52].

The analysis of side effects is fundamental to ensure the vaccination campaign’s effectiveness, especially since the fear of an adverse reaction is one of the main reasons for vaccine refusal [54]. Thanks to these results, future promotion programs and training campaigns among health professionals should aim to overcome behavioral and social factors related to vaccine refusal or hesitancy in the general population [55].

This study, reporting the effectiveness of COVID-19 vaccines in real-world settings, suggests how a vaccination strategy’s objective should also be achieved by prioritizing specific population groups; as part of SARS-CoV-2 epidemic control strategies, the most attention is needed for all healthcare professionals. However, the informal caregivers and healthcare workers outside hospitals were neglected. Moreover, higher vaccination coverage rates are essential to limit the spread of SARS-CoV-2 among patients, but at the same time, the low vaccination coverage reported among older healthcare workers could represent a public health issue because they are a group at increased risk of developing a severe form of COVID-19 [56]. The improvement of the vaccination campaign in this sense could help increase protection and reduce risk exposure to SARS-CoV2 both among healthcare workers and in the general population [21].

Potential limitations of this study are that the different professional categories are not considered and discussed, going beyond the aim of this study; in the same way, information about chronic diseases or other morbidities affecting the healthcare professionals were not collected.

## 5. Conclusions

Overall, the analysis of the hospital-based healthcare workers in this study allows us to focus mainly on re-evaluating the vaccination campaign and prevention policies for designing and delivering a public health service plan efficiently tailored for healthcare professionals.

Disaggregated data alone cannot provide a single answer on the best strategy to adopt for the rollout of COVID-19 vaccinations. A proper understanding of sex and age differences in antibody titer and side effects is the first step toward program-specific preventive strategies and identifying targets. In particular, our data showed that the serological response and the side effects are influenced by sex and age. Indeed, considering the disaggregated data, the antibody titers at different time points highlighted differences between males and females and among different age groups, with a clear decreasing trend in antibody titers in the older age groups. Overall, the serological values were significantly higher in females than in males; in the same way, the reported side effects were more frequent in females than in males. Therefore, intervention policies taking these points into account could help identify a more targeted strategy for the vaccination campaign.

Still, they are essential for identifying priority groups for vaccination against COVID-19 and developing efficient and effective vaccination strategies. However, it can provide insights into some of the most influential decision-making factors according to different scenarios and public health objectives. In the management of COVID-19, considering the sex and gender effects, both direct and indirect, and analyzing the gendered impacts is fundamental for creating effective, equitable policies and interventions.

## Figures and Tables

**Figure 1 vaccines-09-00522-f001:**
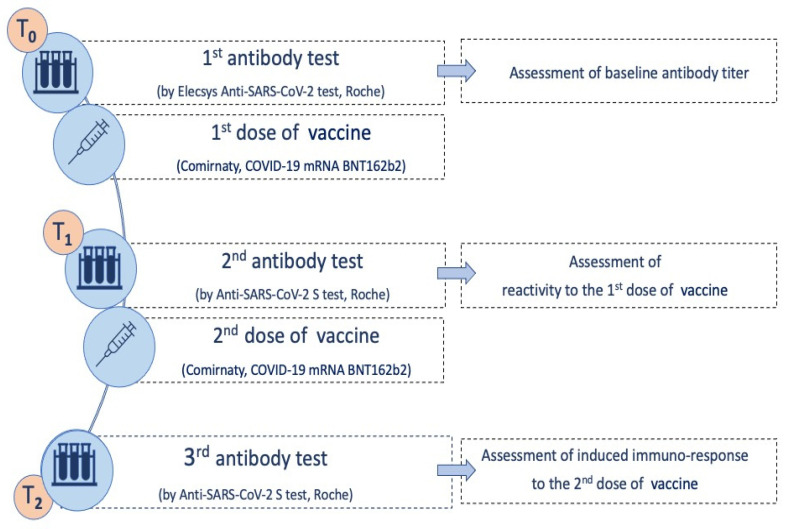
Timing of laboratory evaluation and vaccination. T_0_: blood sampling and serological test to evaluate any presence of SARS-CoV-2 antibodies. T_1_ (21st day): blood sampling and serological test to evaluate the immune response developed following the first vaccine dose. T_2_ (42nd day): blood sampling and serological test to evaluate the immune response developed following the second vaccine dose.

**Figure 2 vaccines-09-00522-f002:**
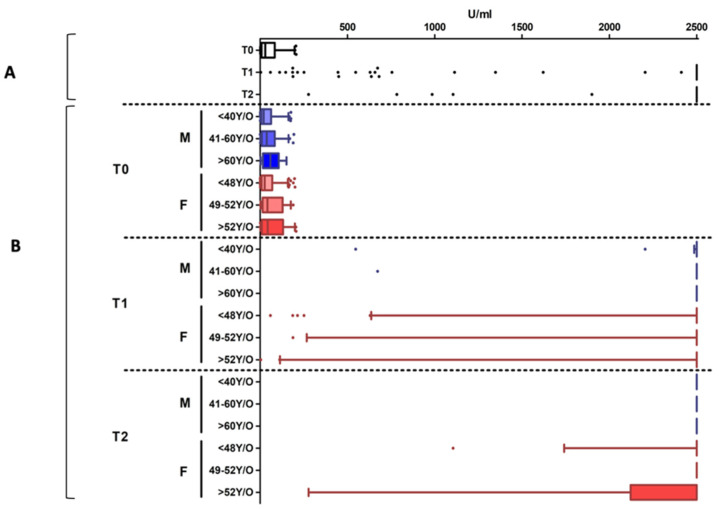
Antibody titers detected in the population with the natural immunity to SARS-CoV-2. Panel (**A**): the data are aggregated. Panel (**B**): disaggregated data by sex (Females: F, in red; Males: M, in blue) and age groups. Individual dots represent outliers. In this population, a high reaction after both vaccine doses is detected, without differences in age and sex groups.

**Figure 3 vaccines-09-00522-f003:**
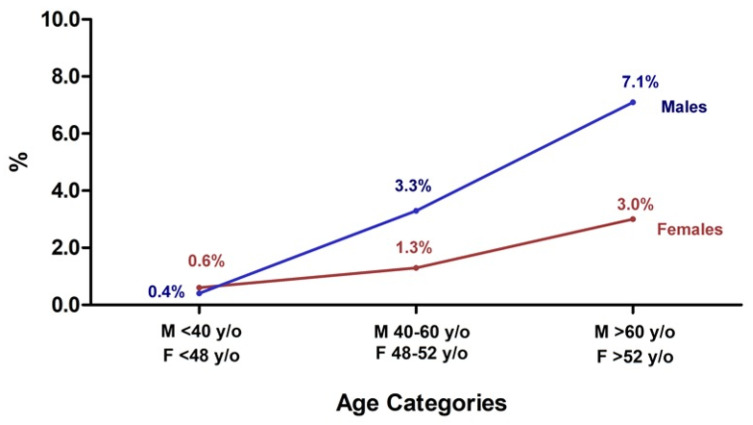
Percentage of nonreactive healthcare workers to the first vaccine dose in males (blue line) and females (red line) and different age groups. An increased percentage of nonreactive subjects was observed from 0.4% to 3.3% and 7.1% in <40 yr, 40–60 yr, and >60 yr, respectively, in males, and in females from 0.6% to 1.3% and 3.0% in <48 yr, 48–52 yr, and >52 yr, respectively.

**Figure 4 vaccines-09-00522-f004:**
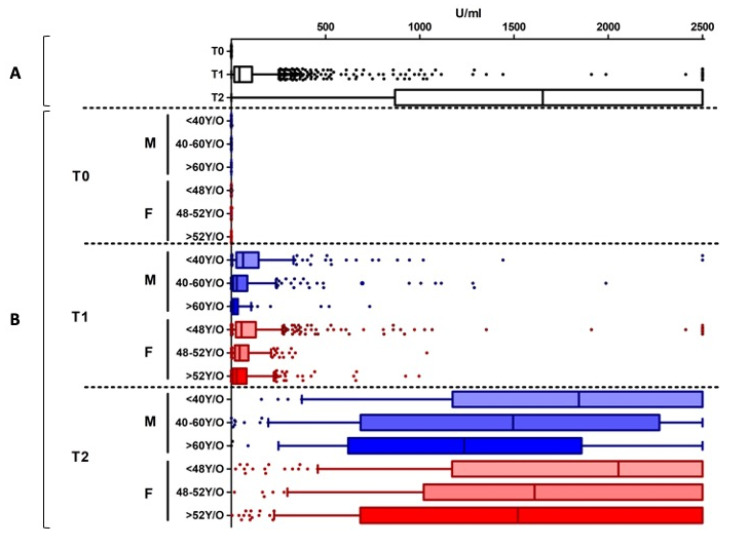
Antibody titers at different time points. Panel (**A**): the data are aggregated. Panel (**B**): disaggregated data by sex and age groups. Disaggregated evaluation allowed highlighting differences between males (M, in blue) and females (F, in red) and among different age groups with a clear decreasing trend in antibody titers in the older age groups, both at T_1_ and T_2_. Overall, the serological values were significantly higher in females than in males. Individual dots represent outliers.

**Figure 5 vaccines-09-00522-f005:**
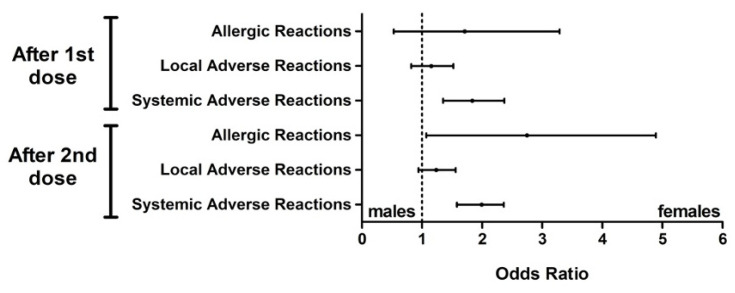
Odds ratios of self-reported adverse reactions after the first and second doses. Local reactions had a similar incidence in males and females after the two doses of vaccine and in the allergic reactions after the first dose; moreover, systemic and allergic reactions after the second dose were more frequent in the females.

**Figure 6 vaccines-09-00522-f006:**
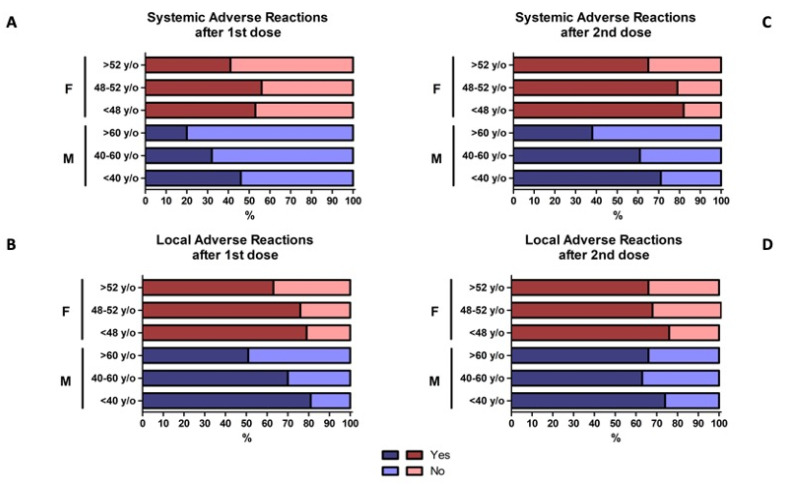
Self-reported adverse reactions after the first (**A**,**B**) and second doses (**C**,**D**) in males, females, and different age groups. Systemic adverse reactions in panels A and C; local adverse reactions in panels B and D. Overall, the second dose of the vaccine was associated with a higher incidence of systemic reactions in females (F, in red) than in males (M, in blue); furthermore, an inversely proportional correlation was found between age and localized reactions in both males and females.

**Table 1 vaccines-09-00522-t001:** The five steps of the Gender Impact Assessment (GIA) method applied to the COVID-19.

Step 1	To define context, objectives and indicators	Context: vaccination campaign against SARS-CoV-2 Objectives: provide gender-based recommendations for vaccination campaign against SARS-CoV-2 Indicators: antibody titer against Sars-Cov-2 before and after the vaccine; side effects due to the vaccine
Step 2	To explicate the relevance for GIA	Gender dynamics: exposure to SARS-CoV-2 and the vaccine access Direct impacts of gender dynamics: access to vaccine Indirect impacts of gender dynamics: intermediate access to vaccine
Step 3	To identify gender impacts	Gender stereotypes: women are considered more suitable for care and assistance jobs Hierarchical positioning: hospital-based healthcare workers vs informal caregivers and healthcare workers outside hospitals Unequal condition: access to vaccine for informal caregivers and healthcare workers outside hospitals vs hospital-based healthcare workers
Step 4	To evaluate gender impacts	Harmful impacts of gender bias: females working in healthcare are most at-risk for being exposed to SARS-CoV-2 infection Aspects that reduce inequalities: to provide vaccine to all individuals at comparable risk in healthcare
Step 5	To provide recommendations for adjustments	Suggestions for reducing inequalities: to include sex-disaggregated reporting of immuno-response and side effects Development of strategies to transform negative impacts of gender-gap into positive ones: healthcare workers could be included in decision-making regarding the campaign against SARS-CoV-2, taking into account knowledge and experience gained by women in the management of COVID-19

## Data Availability

The data presented in this study are available on request from the corresponding author.

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
