# Peer review of "The Gender Impact Assessment among Healthcare Workers in the SARS-CoV-2 Vaccination—An Analysis of Serological Response and Side Effects"

_vaccines, 2021, doi:10.3390/vaccines9050522_

Round 1

Reviewer 1 Report

This work provides information about the immune response and reaction to COVID-19 vaccine among healthcare workers taking into account the differences in terms of sex and age.  This information is interesting but discussion and conclusions Sections should be significantly improved.  

First, the authors should establish a better connection between their findings and their policy recommendations. It is not clear how this paper helps to evaluate the vaccination campaigns. I agree with the need of taking into account the sex but the relevant question is in what way. In the present version of the paper it is not clear whether healthcare women workers should receive COVID-19 vaccine before healthcare men workers, independent of the age and whether vaccine dosage should be different by sex, by age or by both factors. This evidence with respect to healthcare staff of hospitals explained in a more clear way would be useful for authorities . Additionally, the authors cannot give any assessment about informal caregivers and healthcare workers outside hospitals because they are not included into the sample. In the same way, the authors cannot say anything about vaccine refusal since the sample is about people who have received the vaccine. With respect to the side effects, the authors should limit to make clear whether adverse reactions depend on age, on sex or both and hence, to recommend a closer monitoring of the group with higher level of adverse reactions.  

Second, more information about the sample should be provided.  In particular, we do not know the margin error of the results or whether the sample is biased or not. How many health care workers work in this hospital? How is the size of hospital with respect to other hospitals? How is its worker distribution in terms of sex? How is its worker distribution in terms of age? … An appendix with the main descriptive statistics of each sample should be included.

Third, some relevant information is not present. How is the risk of the interviewees? I assume that the risk will be different depending on the professional category or having chronic diseases. This drawback should be mentioned in the conclusions.

Author Response

Point by point

The efficiency of SARS-CoV-2 vaccination campaign exploiting the Gender Impact Assessment among Healthcare Workers by Di Resta C et al.

The Authors thank the Reviewers for the careful work and the valuable suggestions that can improve the quality of the manuscript.

Below a point-by-point response to the issues addressed with specific reference to the changes in the text (in yellow).

We hope that the new version of the manuscript has clarified the weak points highlighted by the Reviewers.

Reviewer 1

This work provides information about the immune response and reaction to the COVID-19 vaccine among healthcare workers taking into account the differences in terms of sex and age.  This information is interesting but discussion and conclusions Sections should be significantly improved.  

First, the authors should establish a better connection between their findings and their policy recommendations. It is not clear how this paper helps to evaluate the vaccination campaigns. I agree with the need of taking into account the sex but the relevant question is in what way. In the present version of the paper it is not clear whether healthcare women workers should receive COVID-19 vaccine before healthcare men workers, independent of the age and whether vaccine dosage should be different by sex, by age or by both factors. This evidence with respect to healthcare staff of hospitals explained in a more clear way would be useful for authorities.

A proper understanding of sex and age differences in antibody titer and side effects is the first step to program-specific preventive strategies and identify targets.

In particular, our data showed that the serological response and the side effects are influenced by sex and age. Indeed, considering the disaggregated data, the antibody titers at different time points highlighted differences between males and females and among different age groups, with a clear decreasing trend in antibody titers in the older age groups. Overall, the serological values were significantly higher in females than in males; in the same way, reported side effects were more frequent in females than in males. Therefore, intervention policies taking these points into account could help identify a more targeted strategy for the vaccination campaign.

Moreover, our data suggest that a single dose of vaccine elicits immune responses in seropositive individuals with post-vaccine antibody titers comparable to or exceed titers found in naïve individuals who received two doses. These observations are in line with the first vaccine dose serving as a boost in naturally infected individuals, providing a rationale for updating vaccine recommendations to consider a single vaccine dose sufficient to reach immunity. A serological test that measures antibodies to the spike protein could be used to screen individuals before vaccination; this "vaccine triage" could be a way to suggest the distribution of vaccine doses based on the antibody titer. Such policies would optimize (in particular due to the scarcity of vaccine doses and to logistic constraints) vaccination campaign by defining vaccine inoculation priorities as a function of antibodies titer. However, it could limit the reactogenicity experienced by natural seropositive individuals.

A sentence has been added in line 365.

Additionally, the authors cannot give any assessment about informal caregivers and healthcare workers outside hospitals because they are not included in the sample.

Our study population encompassed specifically only the professionals inside hospitals, which has been the first workers' category to be included in this vaccination campaign (range time for the first dose from the beginning of January to the middle of February 2021 (line 74).

Desirable measures to optimize the vaccination campaign include targeted initiatives to increase the access of vulnerable groups to healthcare services. Informal caregivers perform similar tasks as nurses and healthcare professionals, but it’s often unclear whether they’ve been prioritized to receive the COVID-19 vaccine. The criteria allowing eligibility to get vaccinated as an informal caregiver differs between States and Countries.

In Italy, it was only in the first half of March 2021 that the priorities in the Covid-19 vaccine plan were updated. Priority is given to the elderly, frail people, severely disabled/disabled people and their cohabiting caregivers and family members, and specific booking channels have been set up. From the data released by the Ministry of Health, as of 05/05/2021 (https://www.governo.it/it/cscovid19/report-vaccini/) more than 21,600,000 administrations have been carried out, of which more than 3,600,000 to the category "Fragile Subjects and Caregivers" (second in order of number only to subjects over 80).

In this study, the use of Gender Impact Assessment has highlighted the inequalities of access to the vaccination campaign for informal caregivers and healthcare workers outside hospitals vs hospital-based healthcare workers. The Authors emphasize this point in line 310.

In the same way, the authors cannot say anything about vaccine refusal since the sample is about people who have received the vaccine. In the discussion, the Authors better explain the comment about the importance of adopting a good strategy and promotion programs in the vaccination campaign to reduce the refusal and hesitancy in the general population.

This study is specifically focused on the healthcare workers involved in the vaccination campaign, which is still ongoing right now. Therefore, the Authors can’t estimate the percentage of refusal in the San Raffaele population. However, in line 346, the importance of a good strategy for the vaccination campaign is discussed to reduce confounding factors that can lead to refusal and hesitancy in the general population.

With respect to the side effects, the authors should limit to make clear whether adverse reactions depend on age, on sex or both and hence, to recommend a closer monitoring of the group with higher level of adverse reactions.

The Authors added a sentence in line 338.

Second, more information about the sample should be provided.

In particular, we do not know the margin error of the results or whether the sample is biased or not.

Until the middle of March, we collected 2486 completed survey, as described in Section 3.3.

In particular, respondents were 19% and 29%, respectively, after the first and the second dose (line 245). More in details, questionnaire replies were received from 22% of the vaccinated females and 14% of vaccinated males after the first dose, and 32% of vaccinated females and 22% vaccinated males after the second dose. Considering the disaggregated data, respondents among different groups of sex and age are distributed as described in Suppl. Table 5 highlights that the younger group filled out the survey more than the elderly (line 249).

How many health care workers work in this hospital? How is its worker distribution in terms of sex? How is its worker distribution in terms of age?

In San Raffaele Hospital, the total number of healthcare professionals is 6395, including the medical residents. The distribution of the entire population is reported in Suppl.Table 1. As described in line 111, the sex groups of the healthcare professionals represented in this study (64,4% females; 35,6% male) reflect the ratio between two groups in the total San Raffaele Healthcare workers population (64% females; 36% males_Suppl.Table 1).

How is the size of hospital with respect to other hospitals?

San Raffaele Hospital is one of the biggest Italian hospitals. The second Institute in Lombardy with more than 60000 treated hospitalized patients and 1 million outpatient visits in one year. This University Hospital is characterized by having 1300 inpatient beds.

An appendix with the main descriptive statistics of each sample should be included.

The Authors described the requested additional information in the Supplementary Tables.

Third, some relevant information is not present. How is the risk of the interviewees? I assume that the risk will be different depending on the professional category or having chronic diseases. This drawback should be mentioned in the conclusions.

The studied population can be considered high-risk workers compared to the general population, given the professional risk itself. Indeed, the frontline physicians and nurses may have a higher risk. However, the different professional categories are not considered and discussed, going beyond the aim of this study. In the same way, information about chronic diseases or other morbidities affecting the healthcare professionals were not collected for this study, which is focused on gender differences (line 110).

We thank the Reviewer for the raised suggestion that could be interesting for a further study.

Reviewer 2 Report

The authors have measured SARS-CoV-2 antibody levels in a population of healthcare workers undergoing COVID-19 immunization and have applied a gender impact assessment.  Furthermore, they have investigated side-effects and additional factors.  The study and data generated are of significant interest.  The English and sentence structure throughout the manuscript could be improved through some minor modifications.

Comments by section

Title.  The title could be improved to better reflect what has been done eg. "A gender impact assessment of SARS-CoV-2 vaccination of a population of Italian healthcare workers - an analysis of antibody responses and side effects".  I would suggest that the efficiency of vaccination hasn't actually being measured.

Abstract.  The abstract describes the reason for the study, the methods, and the conclusions.  It would be useful if a couple of sentences were devoted to the experimental results.  The sentence in lines 20-21 doesn't make sense to me.  The data is of significant interest and it would be a shame for readers to pass it by.

Introduction.  The introduction is comprehensive and referenced adequately.

Methods.  The methods used are adequately described.  Over what time period was the study conducted?

Results.  I suggest that Tables 1 and 2 are replaced by scatterplots or bar charts where medians and inter-quartile ranges can be visualized.  Figure 3 is helpful.

Discussion.  The discussion is comprehensive and appropriately referenced.

Author Response

Point by point

 The authors have measured SARS-CoV-2 antibody levels in a population of healthcare workers undergoing COVID-19 immunization and have applied a gender impact assessment.  Furthermore, they have investigated side-effects and additional factors.  The study and data generated are of significant interest.  Throughout the manuscript, the English and sentence structure could be improved through some minor modifications.

Comments by section

Title.  The title could be improved to better reflect what has been done eg. "A gender impact assessment of SARS-CoV-2 vaccination of a population of Italian healthcare workers - an analysis of antibody responsesand side effects".  I would suggest that the efficiency of vaccination hasn't actually being measured.

As suggested by the Reviewer, we modified the title as: “The Gender Impact Assessment among Healthcare Workers in the SARS-CoV-2 vaccination - an analysis of serological response and side effects.

Abstract.  The abstract describes the reason for the study, the methods, and the conclusions.  It would be useful if a couple of sentences were devoted to the experimental results.  The sentence in lines 20-21 doesn't make sense to me.  The data is of significant interest, and it would be a shame for readers to pass it by.

As suggested by the Reviewer, the Authors added a brief description of the main results into the Abstract, clarifying the sentence in line 20 (line 21-25).

Introduction.  The introduction is comprehensive and referenced adequately.

Methods.  The methods used are adequately described.  Over what time period was the study conducted?

This study is a picture of the first phase of the vaccination campaign in San Raffaele Institute, with a range time for the first dose from the beginning of January to the middle of February 2021. As suggested by the Reviewer, the Authors added this information in line 74.

Results.  I suggest that Tables 1 and 2 are replaced by scatterplots or bar charts where medians and inter-quartile ranges can be visualized.  Figure 3 is helpful.

Table 2 is replaced with scatterplots, as suggested (line 198). Table with the serological values at T0 in the population with the natural immunity to SARS-CoV-2 is reported in Supplementary materials (Suppl. Table 2; line 194 in the text).

Discussion.  The discussion is comprehensive and appropriately referenced.

Round 2

Reviewer 1 Report

Two paragraphs included in the author response should be added to the article -in the discussion Section or in the conclusion Section- :

"Moreover, our data suggest that a single dose of vaccine elicits immune responses in seropositive individuals with post-vaccine antibody titers comparable to or exceed titers found in naïve individuals who received two doses. These observations are in line with the first vaccine dose serving as a boost in naturally infected individuals, providing a rationale for updating vaccine recommendations to consider a single vaccine dose sufficient to reach immunity. A serological test that measures antibodies to the spike protein could be used to screen individuals before vaccination; this "vaccine triage" could be a way to suggest the distribution of vaccine doses based on the antibody titer. Such policies would optimize (in particular due to the scarcity of vaccine doses and to logistic constraints) vaccination campaign by defining vaccine inoculation priorities as a function of antibodies titer. However, it could limit the reactogenicity experienced by natural seropositive individuals"

"the different professional categories are not considered and discussed, going beyond the aim of this study. In the same way, information about chronic diseases or other morbidities affecting the healthcare professionals were not collected for this study" 
